# Optimising a couples-focused intervention to increase couples' HIV testing and counselling using the person-based approach: a qualitative study in Kwa-Zulu Natal, South Africa

Katherine Morton [1], Tembeka Mhlakwaphalwa,[2] Lindani Msimango,[2] Alastair van Heerden,[2,3] Thulani Ngubane,[2] Philip Joseph,[2] Nathi Ngcobo,[2] Z Feng,[4] Victoria Hosegood,[5] Heidi van Rooyen,[2,6] Nuala McGrath[7,8]

For numbered affiliations see end of article.

**Correspondence to**
Katherine Morton;
k.s.morton@soton.ac.uk

## ABSTRACT

**Objectives** This qualitative study explored how to optimise a couples-focused intervention to promote couples HIV testing and counselling (CHTC).

**Setting** Community setting in Kwa-Zulu Natal, South Africa.

**Participants** Qualitative interviews were conducted with 20 couples who had participated in a couples-focused intervention and five staff members delivering the intervention. Partners were interviewed individually by researchers of the same gender.

**Intervention** A couples-focused intervention comprised of two group sessions and four couples counselling sessions was previously shown to significantly increase uptake to CHTC in Kwa-Zulu Natal, South Africa. However, more than half of couples participating in the intervention still chose not to test together during follow-up.

**Analysis** The transcripts were analysed using the table of changes from the person-based approach. Proposed optimisations were discussed with a community group to ensure the intervention was as persuasive and acceptable as possible.

**Results** Many couples found it challenging to discuss CHTC with their partner due to an implied lack of trust. Optimisations to the intervention were identified to increase readiness to discuss CHTC, including education about serodiscordance, discussions about CHTC by peer mentors and open discussion of personal barriers to CHTC during couples' counselling sessions. Additional training for staff in open questioning techniques could help them feel more comfortable to explore couples' perceived barriers to CHTC, rather than advising couples to test. A logic model was developed to show anticipated mechanisms through which the optimised intervention would increase uptake to CHTC, including increasing knowledge, increasing positive outcome beliefs and managing negative emotions.

**Conclusions** In-depth qualitative research informed optimisations to a couples-focused intervention for further evaluation in South Africa to encourage uptake to CHTC. Suggestions are made for optimal methods to gain open feedback on intervention experiences where participants may be reluctant to share negative views.

### Strengths and limitations of this study

► Conducting qualitative interviews with partners separately, in their own homes, and with a researcher of the same gender helped promote rapport and facilitated open reflections about their relationship.

► Interviewing staff members responsible for delivering the intervention in addition to couples participating in the intervention provided a more in-depth understanding of potential barriers to couples HIV testing and counselling.

► The Table of Changes from the Person-Based Approach provided a rigorous method to analyse the qualitative interviews. The use of guiding principles and a logic model ensured that optimisations were grounded in the specific psychosocial context of this population and drew on theoretical constructs.

► Conducting the interviews 5 years after the intervention had been delivered may have limited participants' ability to accurately recall their experiences and perceptions.

► Many participants appeared reluctant to share any negative views of the intervention, and alternative methods such as the use of vignettes may have helped obtain more critical feedback about the couples-focused intervention.

## INTRODUCTION

Couples HIV testing and counselling (CHTC) is recommended by WHO[1] and is now part of South African policy guidelines,[2] however, very few couples in South Africa have tested together.[3 4] This is against a background of continued high levels of HIV prevalence, particularly in KwaZulu-Natal (KZN), South Africa. In a large community-based survey of adults in KZN, 19% of men and 41% of

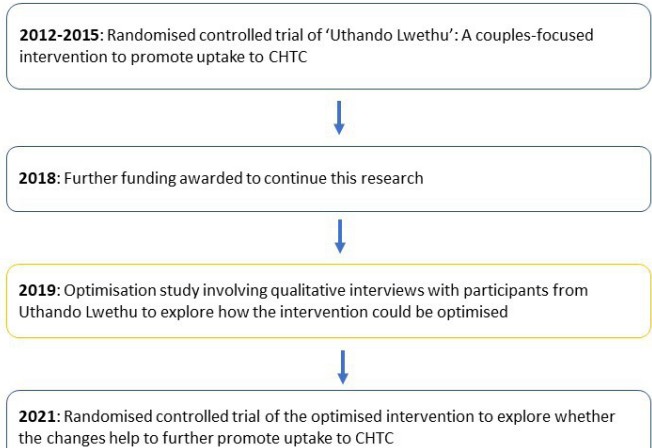

**2012-2015**: Randomised controlled trial of 'Uthando Lwethu': A couples-focused intervention to promote uptake to CHTC

**2018**: Further funding awarded to continue this research

**2019**: Optimisation study involving qualitative interviews with participants from Uthando Lwethu to explore how the intervention could be optimised

**2021**: Randomised controlled trial of the optimised intervention to explore whether the changes help to further promote uptake to CHTC

**Figure 1** Stages of research leading up to and from present optimisation study. CHTC, couples HIV testing and counselling.

women were living with HIV in 2017.[5] The majority of couples in stable sexual relationships reported being unaware of their partner's HIV status (57%),[6] and testing rates remain low, particularly for men.[7] CHTC can contribute to reductions in HIV incidence as a compliment to the universal test and treat policy in South Africa[8] and individual focused testing campaigns.[9]

In 2012–2015, a couples-focused intervention ('Uthando Lwethu') was implemented in a rural community setting in uMgungundlovu, KZN. This community has high levels of migration and unemployment, and low marriages rates, so many couples in the intervention were neither married nor cohabiting.[10] At the outset, nearly 40% of the sample had never had an HIV test, and most had not shared their HIV status with their current partner.[10] The intervention was shown to significantly increase uptake to CHTC, with 42% of couples attending CHTC within a 9-month follow-up period compared with 12% in the control group.[10] Nonetheless, over half of couples receiving the intervention chose not to test within the study period.

In order to optimise the couples-focused intervention for future implementation, we undertook further supplementary research to explore how the intervention content and delivery could be modified to support couples to feel ready for CHTC. This paper presents the findings from a qualitative study with couples and staff involved in the randomised controlled trial (RCT) of the Uthando Lwethu intervention. The person-based approach (PBA)[11] was adopted, which complements evidence and theory in developing and optimising effective interventions and has been used to successfully create behaviour change interventions that are feasible, persuasive, motivating and engaging.[12] This approach recommends conducting in-depth research to develop an understanding of the psychosocial context and underlying beliefs in the target population that could influence engagement with behaviour change. Guiding principles are developed to describe specific behavioural barriers

for a given population and context and identify particular intervention features to overcome these barriers. In addition, this intervention optimisation process was informed by recent evidence regarding facilitators to CHTC,[13–15] and a consideration of the theoretical mechanisms through which the optimised intervention might impact couples' readiness to undertake CHTC.

Our research questions were:
a. How can a couples-focused intervention be optimised to overcome barriers to engaging in CHTC in KZN?
b. What are the theorised mechanisms of action through which the optimised intervention is anticipated to change behaviour?

## METHODS
### Optimisation study design
Figure 1 shows how this optimisation study fitted into a wider body of research, with the orange box showing this study.

This was a qualitative study using retrospective semistructured interviews to explore possible barriers to engaging in CHTC among couples participating in the Uthando Lwethu intervention. Due to the logistics of research funding and contracts, the interviews were conducted approximately 5 years after participants took part in the intervention. We report the optimisation process using the GUIDED (Guidance for reporting intervention development studies in health research) checklist, in line with best practice for reporting intervention development.[16] The COREQ (COnsolidated criteria for REporting Qualitative research) checklist was used for reporting (online supplemental file 1).

### Uthando Lwethu Intervention, prior to optimisation
As this study aimed to describe the optimisation of a couples-focused intervention to promote CHTC, the original content and delivery of the Uthando Lwethu intervention trialled in the 2012–2015 RCT[17] is described here.

The intervention was delivered in a community setting in KZN to 168 couples who had been in a relationship for at least 6 months. Just under two-thirds of participants had had an HIV test before (63% of males and 60% of females) but only 20% of these had told their partner their HIV status. No couples had mutually disclosed their HIV status to each other.[10] The mean age was 26 years and 20% of the couples were married.[10]

Uthando Lwethu was a couples-focused intervention composed of two group sessions and four couples counselling sessions that aimed to increase CHTC uptake. It was developed based on interdependence theory which states that positive relationship dynamics can lead to a transformation of motivation, in which couples are motivated to engage in health behaviours due to a focus on the relationship rather than an individual perspective. It was theorised that couples would be more willing to engage in CHTC if they have greater commitment and trust in their relationship.[17] The intervention was adapted from the

| Table 1 | Uthando Lwethu intervention content |
| --- | --- |
| **Session** | **Content** |
| Group session 1 | A mixed gender half-day session with approximately 15–20 couples. Information provision on TB, HIV, contraception and alcohol. This was designed as a baseline session for all couples before randomisation, and was not not part of the intervention content. Couples were randomised to the intervention or control group at the end of this session. |
| Group session 2 | A single gender half-day session including discussions about relationship dynamics, HIV treatment, gender norms and practical skills sessions on using condoms and learning communication techniques. |
| Couples counselling session 1 | A 90–120 min counselling session including discussion of relationship expectations, communication skills and goal-setting. |
| Couples counselling session 2 | A 90–120 min counselling session to develop communication skills, discuss barriers to achieving their goals and engage in problem-solving, and enhance positive relationship dynamics and intimacy using activities to help focus on what they like about their relationship and their partner. |
| Couples counselling session 3 | A 90–120 min counselling session to continue building communication and problem-solving skills. |
| Couples counselling session 4 | A 90–120 min counselling session to discuss maintenance of any behaviour changes, goal-setting for the future and identify possible challenges they might encounter that could impact their relationship. |

TB, tuberculosis.

Prevention and Relationship Enhancement Programme, which aims to improve relationship dynamics through education in problem-solving and communication skills.[18]

Table 1 provides a summary of the content of each session. The full intervention development process is described elsewhere.[17]

Forty-two per cent of couples in the intervention group attended CHTC, of whom 46% were concordant HIV-negative, 30% were concordant HIV-positive and 24% were serodiscordant.[10] In 54% of the couples who took up CHTC in the intervention group, at least one partner was testing for the first time. Fifty nine percent of the participants who were diagnosed as HIV-positive were new diagnoses.[10]

### Optimisation study participants
Couples were purposively sampled according to their engagement; defined in terms of how many counselling sessions the couple attended, and whether or not they engaged in CHTC as follows:
► Attended CHTC after attending the first couples counselling session.
► Attended CHTC after attending 2–4 couples counselling sessions.
► Attended four couples counselling sessions but did not attend CHTC.
► Attended<4 couples counselling sessions and did not attend CHTC.
► Did not attend any couples counselling sessions, may or may not have attended CHTC.

We sought to sample the same number of couples across the five groups. Couples in group A were anticipated to have the strongest motivation to test together, based on their decision to get tested after only one counselling session, while those in group B appeared to need more support to reach the decision to test together. Those in group C engaged well with the intervention but chose not to test together, while those in group D showed lower engagement with the intervention and decided not to get tested. Group E did not attend any couples counselling sessions, suggesting low motivation. This enabled us to explore diverse perceptions of the intervention and consider why couples chose to test or not to test.

In addition, five staff members who delivered the intervention were interviewed.

### Recruitment procedures
This study took place within a community research site, which facilitated ongoing follow-up of people via contacts in the community, and generally helps ensure service provision. Therefore, recruitment was conducted indirectly by community members who had acted as community advisory board members during the Uthando Lwethu trial. If couples were still in the local area, the community member gave each partner a card inviting them to contact the research team if they were interested in discussing their experiences of the intervention.

The staff members were invited to interview opportunistically based on availability.

### Interview procedures
Semistructured qualitative interviews were conducted face to face in the community by a gender-matched qualitative researcher (TM; female, MSc or LM; male, MSc). TM and LM are research assistants from the local community trained in qualitative interviewing skills by KM. KM is a female health psychologist based in the UK. TM, LM and KM were not involved in the original intervention study. Partners were interviewed separately, and where possible, simultaneously. The interviews took place from August to October 2019, and informed consent was taken at the start of each interview.

Participants' experiences of the two group sessions were explored at the start of the interview. Questions were then

tailored according to whether the couple had attended counselling and/or CHTC, to explore their experiences of these events or to ask how they decided not to attend (online supplemental file 2). Participants were reimbursed ZAR120 (US$8). Interviews were audiorecorded, and researchers made field notes after each interview which were discussed with the team.

The interviews were conducted in isiZulu and translated into English during transcription either by the researcher or a research assistant. TM and LM read the English translation of each interview to ensure they were happy this was an accurate record. Researchers completed a debrief form after each interview to encourage reflection.

The staff interviews were conducted in English and transcribed verbatim. One interview was conducted in person (LM), while the other four were conducted from the UK by video call (KM; female, PhD, employed as a health psychology researcher). Informed consent was taken in advance on-site. The staff interview schedule explored experiences of delivering the intervention (online supplemental file 3).

### Patient and public involvement

During an in-depth half-day workshop, a community working group of eight volunteers from the local community contributed to the interpretation of interview findings and decisions about optimising the intervention. The group included one couple who had taken part in the Uthando Lwethu intervention, one couple who were part of the community working group during the trial, and four individuals who were new to the intervention.

In addition, a participant feedback event was organised in which findings from the qualitative interviews were presented back to couples and staff members who had participated in interviews, and a discussion held about their perceptions of the findings.

For both workshops, we explained how the Uthando Lwethu intervention was theorised to increase uptake to CHTC, to help participants understand the rationale behind the intervention procedures. We then gave an overview of the questions asked during the interviews, to explain the context in which the data were collected. Key quotes were selected from the interviews to demonstrate each barrier to CHTC, and these were displayed on a screen for the groups to discuss and come up with possible solutions, working in smaller groups. We then discussed their ideas as a larger group, presented other possible solutions, and the group were asked to discuss their perceptions of the value and feasibility of each. The workshops were not recorded but note takers were present in each group.

### Analysis

The interviews were analysed by KM using the Table of Changes from the PBA.[19] The Table of Changes provides a technique for rapid inductive qualitative analysis, involving the collation of positive and negative quotes about each aspect of the intervention in order to identify underlying beliefs that could influence engagement with the intervention and target behaviours (ie, CHTC). Using these quotes, KM identified possible optimisations to the intervention to address barriers and promote facilitators to CHTC, drawing on theory, evidence and couples testing guidance.[1] These possible optimisations were discussed in detail with an expert stakeholder group, including all coauthors, Lynae Darbes (health psychologist and couples counsellor) and the community working group. The optimisations were prioritised using the Must have, Should have, Could have, Would like if time permits framework in order to identify those that were essential to promote CHTC.[20] This pragmatic, rapid qualitative analysis approach was consistent with the aim of the study to identify optimisations to the intervention through understanding barriers to the target behaviour and experiences of the intervention.

Guiding principles were developed based on the detailed understanding of barriers to CHTC built up during the qualitative interviews, along with the team's expertise in evidence and theory relating to CHTC. This was an iterative process that aimed to identify key objectives for the intervention to achieve in this context, alongside key features that could be used to achieve these objectives.

A logic model was developed to show how the optimised intervention components were anticipated to influence CHTC uptake. This was a deductive process drawing on existing theory; the optimised intervention components were mapped on to constructs from the Theoretical Domains Framework (TDF)[21] and Capability Opportunity Motivation-Behaviour model.[22] The TDF was chosen as this identified 14 domains from across a wide range of behavioural change theories and uses common terminology to describe approaches to behavioural change.

### RESULTS

A subsample of the 168 couples who took part in the Uthando Lwethu intervention were invited to qualitative interviews (n=36 couples). Thirty-two couples agreed to participate, of whom 20 couples were interviewed. In addition, one couple agreed to participate, but only the female partner was interviewed as the interview with the male partner could not be scheduled. Therefore, in total 41 participants (aged 25–59 years) took part in an individual interview, with four couples interviewed from each of the five target groups. The mean interview duration was 57 min.

The staff members interviewed included the study manager and four staff members who delivered group sessions and couples counselling.

### How can a couples-focused intervention be optimised to overcome barriers to engaging in CHTC in KZN?

Table 2 shows an excerpt from the table of changes analysis.

The following optimisations were identified by the stakeholder group based on the table of changes.

**Table 2** Excerpt from table of changes analysis

| Intervention component | Participant | Positive comments | Negative comments | Possible change | Reason for change Important for behaviour change (IMP); Easy and uncontroversial (EAS); Repeated (REP); Supported by experience from stakeholders or evidence (EXP); Does not contradict (NCON); Not changed (NC) | MoSCoW must do should do could do would like |
|---|---|---|---|---|---|---|
| Single gender session | 4511, Female aged 30–40, group A | It would be girls alone and boys by themselves and you would talk about the problems that you come across with your partner. And you would get solutions for if the problem occurs again, this is how you could face it. Interviewer: Oh. So, how was it? Participant: It was fun because we were comfortable, you could talk comfortably. There was no one with a secret. | | Increase the amount of time spent in single gender sessions to allow more opportunity for open discussion among peers and help couples feel more ready for testing. | IMP, REP | Should have |
| CHTC | 3960, Male aged 30–40, group D | | I think if you check simultaneously with your partner, if there come different results, its better different results come out if you are not testing together and you will have to answer why results are like that and how to solve such a problem if it arises | Fear of serodiscordance needs to be addressed, perhaps through peer mentors, education about serodiscordance. | IMP | Must |

CHTC, couples HIV testing and counselling; MoSCoW, Must have, Should have, Could have, Would like if time permits.

### Individual fears about receiving an HIV positive diagnosis need to be addressed

Fear about how test results will impact on their life was a barrier to CHTC for some participants. One man who did not test during Uthando Lwethu described his fear of getting tested

*'you will never not be scared when testing. You have that fear that you might be HIV positive'* (C4900, Male aged 20–30, Group D).

Another participant who did not test described feeling overwhelmed by the reliance on medication that would be implied by a potentially positive result.

Interviewer: 'What are your concerns about HIV?'

*Participant: 'I think it's the fact that once you become HIV positive you must know that you will be taking medication for the rest of your life. Yes, you can live long but knowing that you will be taking medication for the rest of your life is what concerns me'.* (C3061, Female aged 20–30, Group E).

Based on these barriers, the stakeholder group agreed that the intervention needed to increase confidence that treatment is effective and accessible, and that being HIV positive does not have to mean sickness or losing your status and respect within the community. The group proposed that this education could be delivered by HIV positive peer mentors discussing how treatment helps keep them healthy, as having a role model was theorised to be more effective than information only.

### Couples counselling sessions need to give couples a safe space to discuss perceived barriers to CHTC

The interviews with couples counsellors suggested that their investment in positive outcomes for the couple could lead them to provide advice and try to persuade couples to engage in CHTC.

*'you speak and speak and speak and encouraging them about honesty and trust and everything but still tomorrow, no, I'm not ready to do the testing… I kept on encouraging them that most of the people are testing negative, just to make them think about it'* (Staff member 1, counsellor and facilitator)

Some participants also seemed to expect that the counselling session would involve advice:

*'To get a chance with a third person who is going to advise you on how things should happen'.* (C680, Male aged 30–40, Group B)

The stakeholder group proposed that training facilitators in the rationale for using open questioning could help encourage them to use this technique more often, and thereby enable a couple to discuss why they are not ready to test together and to come up with their own solutions. Open questioning involves asking the couple open-ended questions, such as 'What makes it harder for you to test together?', to enable the couple to discuss their barriers. Training in this technique would encourage facilitators that it is not their responsibility to ensure couples engage in CHTC, and that their success as facilitators is not evaluated based on CHTC uptake. The Healthy Conversations Skills Training package[23] was identified as an appropriate tool to deliver this message during facilitator training. This emphasises to facilitators that their role is to help people recognise their own barriers and to ask open discovery questions to facilitate them in identifying solutions.

### Couples need to feel ready to have conversations about CHTC outside the intervention setting

Couples in the Uthando Lwethu intervention were encouraged to discuss HIV testing with one another outside the intervention setting. However, a common barrier was that discussing HIV testing outside the supportive environment of the intervention was difficult due to the implied lack of trust.

*'Yes, we would talk about it [testing]. He said that it means that I do not trust him, why else would I talk about this thing. You would see from the way he responds that you should not go any further with it.'* (C0011, female aged 20–30 years, Group D)

This contrasts with a quote from a participant who tested with his partner after only one counselling session, describing their rationale for testing together:

*'If she is going to come back already tested while I have not tested, there might be finger-pointing, but if we find out at the same time, we might get the same shock and be able to comfort*

**Table 3** Guiding principles to inform the optimisation of the intervention

| Design objectives | Key intervention features |
| --- | --- |
| Help couples feel close to each other, recognising the positive aspects of their relationship and providing motivation to look after their health together. | Activities to promote focus on the positive aspects of the relationship. |
| Encourage effective communication to address fears and outcome expectancies about testing. Address negative emotions. | 1. Ask open questions to enable couples to identify their barriers to CHTC and come up with their own solutions. 2. Provide effective communication skills training to help couples explore their perceived barriers to testing together. |
| Increase understanding about risk of HIV transmission, serodiscordance and effectiveness of treatment. | 1. Interactive single-gender group activities 2. Open dialogue facilitated by a peer mentor with experience of couples HIV testing 3. Provide standard guidance on HIV prevention to reduce risk of transmission, for example, condom demonstration. |

*each other, and then look for a way forward*'. (C4510, male aged 40–50 years, Group A)

Concerns about trust and blame were reflected in the staff interviews too.

'*I think there's a lot of barriers when in a couple there's STI because there's a lot of blame. There's a lot of feeling that someone is having an affair so people they don't…I think they won't be comfortable*' (Staff member 4, counsellor and facilitator)

The stakeholder group identified several optimisations to the intervention which could facilitate couples to feel ready to discuss CHTC without blame. These included education about serodiscordance to overcome perceptions that infection automatically indicates infidelity, and a case study of a serodiscordant couple was identified as an effective means to convey this message. In addition, it was agreed that inviting peer mentors from the local community to single-gender sessions could help provide information from a credible, relatable source and facilitate a dialogue about barriers to couples testing.

As well as optimising the content of the intervention, the group also proposed optimising the structure to facilitate couples to feel ready to test together. The original intervention involved one single-gender group session, which was highly valued by men and women as a chance to talk openly about relationship issues.

'*Do you know how difficult it is to talk when your boyfriend is here? You cannot be open to talk about whatever, but when you are with other women you can talk about whatever concerning him*' (C0011, Female aged 20–30 years, Group D)

Staff members also noticed that participants were more willing to have open discussions during the single-gender group session. Therefore, the stakeholder group agreed that the total time for participants in single-gender sessions could be increased in order to facilitate an open dialogue around barriers to HIV testing and associated implications at an earlier point in the intervention. This was theorised to help couples feel more ready to discuss testing together outside the intervention.

## What are the theorised mechanisms of action through which the optimised intervention can change behaviour?

In order to inform the process of optimising the intervention, guiding principles were developed based on these findings regarding the main barriers to CHTC, see table 3. These emphasise that the intervention needs to increase intimacy between couples, encourage effective communication about CHTC, and increase understanding.

A logic model was developed to show how this optimised intervention is theorised to promote CHTC uptake (figure 2). Key differences from Uthando Lwethu are shown in red. As in Uthando Lwethu, transforming motivations remains a key mechanism through which the intervention seeks to increase CHTC uptake. This construct comes from communal coping theory,[24] which proposes that by improving relationship functioning and communication within a couple, motivation for looking after health can be shifted from the individual to the couple. In Uthando Lwethu, this approach informed the couples counselling sessions which aimed to improve relationship functioning and communication skills, thereby facilitating difficult discussions that partners may wish to have and indirectly influencing CHTC uptake.

The optimised intervention still seeks to promote transformation of motivation among couples, but alongside this it has a more direct focus on increasing understanding of the consequences of testing, changing outcome expectancies, and managing negative emotions around HIV testing. This was informed by the qualitative interviews which suggested some people needed further support to feel ready to test for HIV. Therefore, the optimised intervention is theorised to operate both through an indirect pathway of improving relationship functioning and intimacy, but also through a more direct pathway of overcoming barriers to CHTC.

## DISCUSSION

This study adopted the PBA to explore couples' and facilitators' experiences and perceptions of an intervention to promote CHTC uptake in KZN, South Africa.

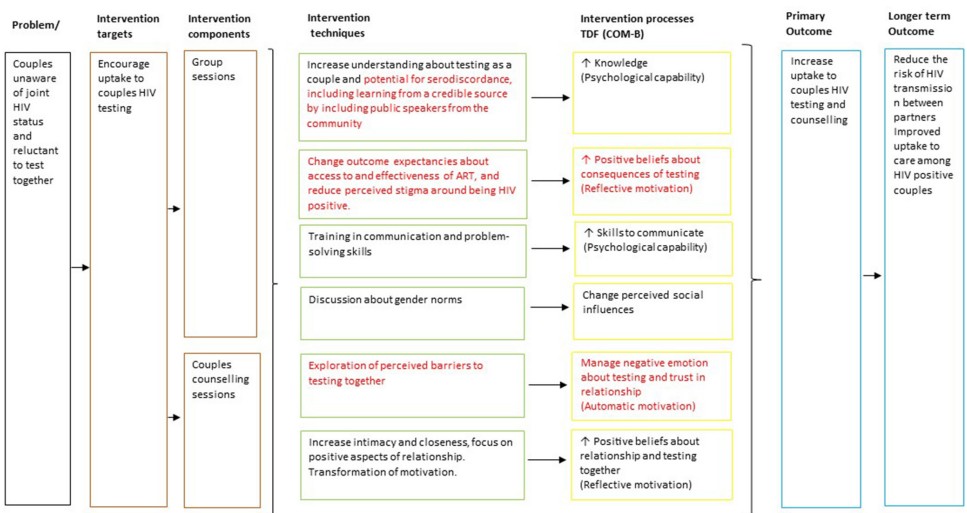

**Figure 2** Logic model.

Suggested optimisations identified with a stakeholder and community group to promote CHTC included group discussion facilitated by peer mentors to help improve outcome expectancies about living with HIV and address concerns about the impact on relationships, a more explicit focus on discussing barriers to CHTC during couples' counselling sessions, and more time in single-gender sessions in order to facilitate open discussion about HIV.

The use of rigorous qualitative methods was a strength of this study as it provided in-depth understanding of facilitators and barriers to couples' HIV testing, which informed optimisation of the intervention components. This is in line with recommendations that qualitative research be used to explore how HIV interventions might change behaviour to help enhance effectiveness.[25] A significant limitation was that the interviews were conducted approximately 5 years after the intervention was delivered. While qualitative interviews always rely on the reconstruction of experiences through the participant's narrative lens,[26] the significant time gap meant that the experiences shared with us were particularly subject to the influence of subsequent events and the passing of years, and should be interpreted carefully. The discussion with the community group helped confirm that the findings are relevant. In an ongoing trial of the optimised intervention, process interviews will be conducted to assist in understanding barriers and facilitators to CHTC at the time couples are making this decision.

While participants openly discussed perceived barriers to CHTC, very few responded to questions about aspects of the intervention they disliked. The use of vignettes about a fictional person may have helped participants feel comfortable to provide more open feedback.

Our finding that couples can find it difficult to discuss CHTC without connotations of blame is consistent with qualitative research in Sub-Saharan Africa showing that people overestimate the likelihood of becoming infected after a one-off sexual encounter and are therefore reluctant to test due to concerns that the test will reveal they have been unfaithful.[14] The perception that discordancy is related to infidelity has been theorised to be exacerbated by social norms for multiple partners; low marriage rates; campaigns that associate HIV prevention with monogamy; and lack of knowledge about the potential for long-term differences in HIV status within couples represented by serodiscordance.[13]

Misunderstandings about serodiscordance and the associated blame presents a significant challenge to implementing CHTC, and supports the need for a supportive environment for couples to discuss HIV testing, with a focus on ensuring the best future together rather than considering past infidelities, as well as education about serodiscordance. This provided a strong rationale for the theoretical shift in the logic model towards addressing perceived barriers more directly, by increasing knowledge of HIV infection and possible outcomes of testing, and exploring negative emotions around CHTC. This is in line with Protection Motivation Theory,[27] in that the optimised intervention seeks to influence couples' threat perceptions and increase their self-efficacy for coping with the threat. However, unlike many interventions which aim to raise perceived risk to change behaviour, this intervention seeks to reduce the perceived risk that CHTC will threaten a couple's trust. As well as being grounded in theory, the optimisations proposed by this study are evidence based as effective strategies for facilitating CHTC uptake, including understanding the benefits of CHTC, and inviting peer mentors to facilitate discussions.[28 29]

The study offered novel insights into the experiences of lay counsellors from the community delivering a couples-focused intervention for HIV testing. As asking open questions is known to be effective to support behaviour change, while telling or giving advice can lead to disagreement from the recipient,[30] the findings from this study suggested that additional training for counsellors to support open questioning and absolve personal responsibility for couples' decisions may be beneficial.

The optimised intervention will be evaluated in a study in KZN, and CHTC uptake will be compared with uptake in the intervention arm of the original RCT. The process of optimisation could be further streamlined in future research by adopting a flexible, adaptive trial design that allows modifications to be made in line with ongoing qualitative process research, such that barriers can be addressed as they come to light.[31]

## CONCLUSIONS

In-depth qualitative interviews identified key barriers to engaging with CHTC, including concerns about trust in the relationship, and a lack of open discussion about barriers to CHTC during counselling sessions. This informed proposed optimisations to an existing couples-focused intervention in line with evidence and theory to overcome these barriers, including the addition of a discussion with peer mentors, education about serodiscordance and training facilitators to use open questions to explore couples' barriers to CHTC. The use of the PBA to intervention optimisation ensured that these changes were grounded in participants' experiences and psychosocial context.

**Author affiliations**
[1]Psychology, University of Southampton, Southampton, UK
[2]Human Sciences Research Council, Sweetwaters, Pietermaritzburg, KwaZulu-Natal, South Africa
[3]School of Clinical Medicine, University of the Witwatersrand Faculty of Health Sciences, Johannesburg, Gauteng, South Africa
[4]Primary Care and Population Sciences, University of Southampton, Southampton, UK
[5]Department of Social Statistics and Demography, Faculty of Social Sciences, University of Southampton, Southampton, UK
[6]University of the Witwatersrand, Johannesburg-Braamfontein, Gauteng, South Africa
[7]School of Primary Care, Population Sciences and Medical Education, Faculty of Medicine, University of Southampton, Southampton, UK
[8]Department of Social Statistics & Demography, Faculty of Social Sciences, University of Southampton, Southampton, UK

**Acknowledgements** Thank you to all participants in this research study. The following people have provided helpful comments or suggestions, or have otherwise enabled this work: Dr Tawanda Makusha (HSRC), and Professor Lynae Darbes (University of Michigan).

**Contributors** LM and TM conducted the interviews. KM trained the interviewers, conducted some staff interviews, analysed the data and wrote the manuscript. AvH, TN, PJ, VH, HvR and NM contributed to the original intervention design and conducted the 'Uthando Lwethu' trial, as well as planning and organising the present qualitative study. NN coordinated the participant recruitment, data collection and community working group and participant feedback events. ZF contributed to intervention optimisation. All authors contributed to decisions about intervention optimisations and reviewed the manuscript.

**Funding** This report is independent research supported by the National Institute for Health Research using Official Development Assistance (ODA) funding (NIHR Global Health Research Professorship, NM, RP-2017-08-ST2-008). KM, TM, LM, NN, ZF and NM were supported by this funding.

**Disclaimer** The views expressed in this publication are those of the author(s) and not necessarily those of the NHS, the National Institute for Health Research or the Department of Health and Social Care.

**Competing interests** The authors have no competing interests to declare.

**Patient consent for publication** Not applicable.

**Ethics approval** Approval was granted by the Human Sciences Research Council (REC 2/19/10/11b) and the University of Southampton (reference number: ERGO 48702) ethics committees.

**Provenance and peer review** Not commissioned; externally peer reviewed.

**Data availability statement** The interview transcripts will not be made available to protect participants' anonymity. Requests for data sharing can be sent to the corresponding author. Full transcripts of interviews are not available to protect participants' anonymity.

**ORCID iD**
Katherine Morton http://orcid.org/0000-0002-6674-0314

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
