## [Reviewer comments · BMJ Open]

ARTICLE DETAILS

TITLE (PROVISIONAL)	Optimising a couples-focused intervention to increase couples' HIV testing and counselling using the person-based approach: A qualitative study in Kwa-Zulu Natal, South Africa.
AUTHORS	Morton, Katherine; Mhlakwaphalwa, Tembeka; Msimango, Lindani; van Heerden, Alastair; Ngubane, Thulani; Joseph, Philip; Ngcobo, Nathi; Feng, Z; Hosegood, Victoria; van Rooyen, Heidi; McGrath, Nuala

VERSION 1 – REVIEW

REVIEWER	LeBlanc, Natalie University of Rochester
REVIEW RETURNED	30-Dec-2020

GENERAL COMMENTS	Great contribution to the literature. Some revisions are in order: There is more description needed in the qualitative analytic approach - was this a qual description? There is a need to review the grammar throughout More description is needed on how the target groups for the study were similar/differ and the implications - and more discussion on ways forward.
--

REVIEWER	Skinner, Donald Human Sciences Research Council, Faculty of Health Sciences
REVIEW RETURNED	06-Feb-2021

GENERAL COMMENTS	Review Optimising a couples-focused intervention to increase couples' HIV testing and counselling using the personbased approach: A qualitative study in Kwa-Zulu Natal, South Africa. bmjopen-2020-047408 The paper is interesting and covers an important area for intervention and knowledge development. However, there are a number of gaps and some questions that the authors need to answer before it can be published. The paper is generally well written with few typos or errors noted. Need to standardize how headings are done, e.g. 3.1.3 and 3.1.4. Sometimes the heading numbers include a heading, while others just went into the paragraph itself, e.g. 3.1.5. Introduction
--

	Good. Much of the content about the service and its role is presented in the methods as part of the description of the intervention. Some description of the community and cultural context in which the intervention took place needs to be included. This may have been in other papers, but the reader needs to have a brief introduction here. Methodology Strong design. The links to the trial need to be better outlined. In particular why did the interviews take place 5 years after the counselling session. As I comment later, this is a major limitation. As this is a qualitative paper, the authors need to introduce themselves in the text. The information provided on pg 9 and 10 is inadequate. Could not find table 1 in the submission. Results Where did the additional women come from in terms of recruitment? This was not raised in the recruitment description. Also, I assume that group B related to the structure of the intervention trial, but needs to be contextualized here or just drop the group identity. The presentation of the barriers is very short. This could be expanded, especially by greater commentary from the authors. The pieces that presented are dominated by quotes, which are good, but the analysis needs to be done by the authors. These should especially contextualize the barriers in the local context of the nature of the community and culture. The solutions that follow the problem have a good logic to them. However, given the overall design, I assume that several the solutions arose from the study population. This would be useful to describe. The quotes are also limited to the barriers. Discussion The discussion needs to be grounded more in the context of the people and additional literature. The period of 5 years between the intervention would make reflection of the intervention very difficult. Even if they had memories to talk about these would have changed so much over the years that little constructive information could have been obtained. Why did the study wait 5 years before doing the interviews? This needs to be explained. I assume it was due to the requirements of the trial, but this needs to be explained. This discounted many of the questions in the schedules. The actual discussion of the findings and how the overall model adjusts could be increased. Much of the space is used for the discussion of strengths, limitations and future study.
--	---

VERSION 1 – AUTHOR RESPONSE

Section	Comment	By	How it was addressed
Methods	There is more description needed in the qualitative analytic approach - was this a qual description?	Reviewer 1	The Table of Changes is a form of qualitative analysis, but it is more rapid and pragmatic than an approach like thematic analysis. We have expanded the description of the Table of Changes analysis and added a sentence to explain the rationale for using this approach (page 11-12): “The interviews were analysed by KM using the Table of Changes from the PBA (Bradbury 2018). The Table of Changes provides a technique for rapid qualitative analysis involving the collation of positive and negative quotes about each aspect of the intervention, in order to identify underlying beliefs that could influence engagement with the intervention and target behaviours (i.e. CHTC). Using these quotes, KM identified possible optimisations to the intervention to address barriers and promote facilitators to CHTC, drawing on theory, evidence, and couples testing guidance (1). These possible optimisations were discussed in detail with an expert stakeholder group, including all co-authors, Lynae Darbes (health psychologist and couples counsellor) and the community working group. The optimisations were

			prioritised using the MoSCoW framework (Must have, Should have, Could have, Would like if time permits) in order to identify those that were essential to promote CHTC (18). This pragmatic, rapid qualitative analysis approach was consistent with the aim of the study to identify optimisations to the intervention through understanding barriers to the target behaviour and experiences of the intervention” In addition, we have added an excerpt from the Table of Changes to the results (Table 2) to help demonstrate how the analysis worked.
All	There is a need to review the grammar throughout	Reviewer 1	The grammar has now been checked throughout.
Methods/discussion	More description is needed on how the target groups for the study were similar/differ and the implications - and more discussion on ways forward.	Reviewer 1	Additional description has been added about the differences between the five study groups (Page 9): “Couples in Group A were anticipated to have the strongest motivation to test together, based on their decision to get tested after only one counselling session, while those in Group B appeared to need more support to reach the decision to test together. Those in Group C engaged well with the intervention but chose not to test together, while those in Group D showed lower engagement with the intervention and decided not to get

			tested. Group E did not attend any couples counselling sessions, suggesting low motivation”. We found it helpful when interpreting quotes about the intervention to consider whether the couple tested together, and how engaged they were in the intervention. This is why we have reported which group a participant in each time we interpret a quote in the results.
Headings	Need to standardize how headings are done, e.g. 3.1.3 and 3.1.4. Sometimes the heading numbers include a heading, while others just went into the paragraph itself, e.g. 3.1.5.	Reviewer 2	Apologies, each heading number now also include a text heading.
Introduction	Good. Much of the content about the service and its role is presented in the methods as part of the description of the intervention. Some description of the community and cultural context in which the intervention took place needs to be included. This may have been in other papers, but the reader needs to have a brief introduction here.	Reviewer 2	Further information about the community and cultural context in which the intervention took place has been added to the introduction (Page 6): In 2012-2015, a couples-focused intervention (‘Uthando Lwethu’) was implemented in a rural community setting in uMgungundlovu, KZN. This community has high levels of migration and unemployment, and low marriages rates, so many couples in the intervention were neither married nor cohabiting (10). At the outset, nearly 40% of the sample had never

			had an HIV test, and most had not shared their HIV status with their current partner (10).
Methods	Strong design. The links to the trial need to better outlined. In particular why did the interviews take place 5 years after the counselling session. As I comment later, this is a major limitation.	Reviewer 2	Figure 1 has been added to better illustrate how this study fits with both what has gone before, and what is yet to come. In addition, a sentence has been added to section 2.1 to explain upfront why this delay occurred (Page 8): “Due to logistics of research funding and contracts, the interviews were conducted approximately five years after participants took part in the intervention. “ We agree this is a significant limitation of the research, and we have added further reflection to our discussion about this (Page 22): “A significant limitation was that the interviews were conducted approximately five years after the intervention was delivered. Whilst qualitative interviews always rely on the reconstruction of experiences through the participant’s narrative lens (22), the significant time gap meant that the experiences shared with us were particularly subject to the influence of subsequent events and the passing of years, and should be interpreted carefully. In

			an ongoing trial of the optimised intervention, process interviews will be conducted to assist in understanding barriers and facilitators to CHTC at the time couples are making this decision”.
Methods	As this is a qualitative paper, the authors need to introduce themselves in the text. The information provided on pg 9 and 10 is inadequate.	Reviewer 2	Additional demographic details about the interviewers and those analysing the data are now included in the paper (Page 10): “TM and LM are research assistants from the local community trained in qualitative interviewing skills by KM. KM is a female health psychologist based in the UK. TM, LM and KM were not involved in the original intervention study”.
Methods	Could not find table 1 in the submission.	Reviewer 2	Many apologies for this omission, Table 1 has now been added.
Results	Where did the additional women come from in terms of recruitment? This was not raised in the recruitment description.	Reviewer 2	Additional detail has been added to explain where the additional woman came from during recruitment (Page 13): “A sub-sample of the 168 couples who took part in the Uthando Lwethu intervention were invited to qualitative interviews (n=36 couples). Thirty-two couples agreed to participate, of whom 20 couples were interviewed. In addition, one couple agreed to participate, but only the female partner was

			interviewed as the interview with the male partner could not be scheduled. Therefore, in total forty-one participants (aged 25-59 years) took part in an individual interview, with four couples interviewed from each of the five target groups”.
Results	Also, I assume that group B related to the structure of the intervention trial, but needs to be contextualized here or just drop the group identity.	Reviewer 2	Agreed, we have removed the reference to Group B in relation to the additional female participant.
Results	The presentation of the barriers is very short. This could be expanded, especially by greater commentary from the authors. The pieces that presented are dominated by quotes, which are good, but the analysis needs to be done by the authors. These should especially contextualize the barriers in the local context of the nature of the community and culture.	Reviewer 2	We agree that the discussion of barriers was brief, however, we would add that this study did not aim to generate detailed insights into couples’ perceptions about testing – which has already been explored in detail elsewhere. Instead the focus was on understanding perceptions about this intervention in particular, and identifying how best to optimise it. The Table of Changes is a rapid, pragmatic analysis method which facilitates this focus on optimising interventions. In line with this, we decided that the first research question which focused explicitly on identifying barriers to CHTC was misleading, and therefore we have now combined our first two research questions to emphasise the focus on identifying

			optimisations to promote CHTC rather than developing an in-depth understanding of perceptions of CHTC in general. This should orientate the reader towards expecting a more pragmatic intervention-focused analysis (Page 7): “How can a couples-focused intervention be optimised to overcome barriers to engaging in CHTC in KwaZulu-Natal?” In addition, we have developed the findings section to add to the description of the barriers and show more clearly how this fed into recommended optimisations (Pages 17-21),
Results	The solutions that follow the problem have a good logic to them. However, given the overall design, I assume that several the solutions arose from the study population. This would be useful to describe. The quotes are also limited to the barriers.	Reviewer 2	We apologise that it was not clear how the solutions or optimisations to the intervention were arrived at. We have now made this clear in the analysis section: “Using these quotes, KM identified possible optimisations to the intervention to address barriers and promote facilitators to CHTC, drawing on theory, evidence, and couples testing guidance (1). These possible optimisations were discussed in detail with an expert stakeholder group, including all co-authors, Lynae Darbes (health psychologist and couples counsellor) and the community working group. The optimisations were

			prioritised using the MoSCoW framework (Must have, Should have, Could have, Would like if time permits) in order to identify those that were essential to promote CHTC (19). This pragmatic, rapid qualitative analysis approach was consistent with the aim of the study to identify optimisations to the intervention through understanding barriers to the target behaviour and experiences of the intervention”. There are also now quotes to show facilitators too: “ It would be girls alone and boys by themselves and you would talk about the problems that you come across with your partner. And you would get solutions for if the problem occurs again, this is how you could face it. Interviewer: Oh. So, how was it? Participant: It was fun because we were comfortable, you could talk comfortably. There was no one with a secret.” (See Table 2) “If she is going to come back already tested while I have not tested, there might be finger pointing, but if we find out at the same time, we might get the same shock and be able to comfort each other, and then look for a way forward’. (4510, male aged 40-50, Group A)” (see findings section 3.1.3)
--	--	--	---

			“when you are with other women you can talk about whatever concerning him” (C0011, Female aged <30 years, group D) (see findings section 3.1.3)
Discussion	The discussion needs to be grounded more in the context of the people and additional literature.	Reviewer 2	Additional literature has been added to the discussion to better ground the findings in the local context (Page 23): “Our finding that couples can find it difficult to discuss CHTC without connotations of blame is consistent with qualitative research in Sub-Saharan Africa showing that people overestimate the likelihood of becoming infected after a one-off sexual encounter and are therefore reluctant to test due to concerns that the test will reveal they have been unfaithful (13). The perception that discordancy is related to infidelity has been theorised to be exacerbated by social norms for multiple partners; low marriage rates; campaigns that associate HIV prevention with monogamy; and lack of knowledge about the potential for long-term differences in HIV status within couples represented by serodiscordance”.
Discussion	The period of 5 years between the	Reviewer 2	As described above, the 5-year gap was due to delays in

	intervention would make reflection of the intervention very difficult. Even if they had memories to talk about these would have changed so much over the years that little constructive information could have been obtained. Why did the study wait 5 years before doing the interviews? This needs to be explained. I assume it was due to the requirements of the trial, but this needs to be explained. This discounted many of the questions in the schedules.		additional funding to continue this research. We agree that this needed further acknowledgement in the discussion and have expanded on this limitation (Page 22): “A significant limitation was that the interviews were conducted approximately five years after the intervention was delivered. Whilst qualitative interviews always rely on the reconstruction of experiences through the participant’s narrative lens (22), the significant time gap meant that the experiences shared with us were particularly subject to the influence of subsequent events and the passing of years, and should be interpreted carefully. In an ongoing trial of the optimised intervention, process interviews will be conducted to assist in understanding barriers and facilitators to CHTC at the time couples are making this decision”.
Discussion	The actual discussion of the findings and how the overall model adjusts could be increased. Much of the space is used for the discussion of strengths, limitations and future study.	Reviewer 2	We agree that the strengths and limitations section was taking up a lot of the discussion, so we have cut this down a little to allow more discussion of the findings and revised logic model while remaining within the word count (Page 23): “Our finding that couples can find it difficult to discuss CHTC without connotations of blame

			is consistent with qualitative research in Sub-Saharan Africa showing that people overestimate the likelihood of becoming infected after a one-off sexual encounter and are therefore reluctant to test due to concerns that the test will reveal they have been unfaithful (13). The perception that discordancy is related to infidelity has been theorised to be exacerbated by social norms for multiple partners; low marriage rates; campaigns that associate HIV prevention with monogamy; and lack of knowledge about the potential for long-term differences in HIV status within couples represented by serodiscordance (12). Misunderstandings about serodiscordance and the associated blame presents a significant challenge to implementing CHTC, and supports the need for a supportive environment for couples to discuss HIV testing, with a focus on ensuring the best future together rather than considering past infidelities, as well as education about serodiscordance. This provided a strong rationale for the theoretical shift in the logic model towards addressing perceived barriers more directly, by increasing knowledge of HIV infection and possible outcomes of testing, and exploring negative
--	--	--	--

			emotions around CHTC. This is in line with Protection Motivation Theory (24), in that the optimised intervention seeks to influence couples' threat perceptions and increase their self-efficacy for coping with the threat. However, unlike many interventions which aim to raise perceived risk to change behaviour, this intervention seeks to reduce the perceived risk that CHTC will threaten a couple's trust. As well as being grounded in theory, the optimisations proposed by this study are evidence-based as effective strategies for facilitating CHTC uptake, including understanding the benefits of CHTC, and inviting peer mentors to facilitate discussions (25, 26)"
--	--	--	---

VERSION 2 – REVIEW

REVIEWER	LeBlanc, Natalie University of Rochester
REVIEW RETURNED	18-Apr-2021

GENERAL COMMENTS	Nice paper. Efforts to improve our understanding as to why CHTC has not gained greater traction in real world settings is warranted. Some considerations to improving the manuscript is as follows: Pg. 8 lines 28-44: In providing insight to the intervention and to the setting the authors provides some information to the population which is helpful but seems to suggest that CHTC is for married couples. However the cohabiting piece seems to be important. What is the cultural expectation regarding "marriage and relationships, are the authors using a western lens exclusively? Additionally there is a need to clarity on who has test and proportion of those who tested, who were diagnosed and who did not disclose. Regarding the intervention outcome, what proportion was testing for the first time.
---

	An explanation of open questioning as a strategy could be described a bit more. There appears to have been several frameworks used and as written is a bit confusing as to the rationale and so some clarity around this upfront is warranted. The conclusion rounds out that more training is needed for counselors but the findings were almost exclusive to the couples. Were there any findings pertaining to the providers that can be shown in support of this. If so this could be incorporated. Grammar and spelling needs to be assessed throughout.
--	---

REVIEWER	Skinner, Donald Human Sciences Research Council, Faculty of Health Sciences
REVIEW RETURNED	19-Apr-2021

GENERAL COMMENTS	Review Optimising a couples-focused intervention to increase couples' HIV testing and counselling using the person-based approach: A qualitative study in Kwa-Zulu Natal, South Africa. bmjopen-2020-047408.R1 Overall, the paper is very interesting. I particularly liked the use of the stakeholder group in the analysis. It also covers an important content area around the content and approach that should be used in counselling. I would recommend publication if the following comments are addressed. I do remain concerned about the 5 year gap, but the study does appear to have achieved sound results that resonate with the community group. Required changes It should be clarified that this study takes place within a community research site. This facilitates the ongoing follow-up of people and ensures service provision. Some greater context has been added, but could still do with more contextualisation. This is a qualitative and partly PAR paper, so context is essential. The method of sampling described is not purposive. It follows the outcomes of the study. Purposive implies a selection based in advance. Or is the purposive sampling based on a distribution across the 5 groups. This is not clear. Okay it is clarified later in the results. A line here in the methods making this clear would facilitate the reading of the manuscript. Pg 9 In 51 When couples were still in should read If couples The introduction of the authors background is still inadequate as it only cover the interviewers. All authors that are part of paper and analysis need to be introduced. The systems used to involve the stakeholders could be unpacked further. How was the analysis presented? Were any tools used to facilitate discussion? This is a novel system, so readers would benefit from hearing more. It would also allow for the methodology to be more easily replicated in other contexts.
---

	Pg 15 ln 24 I am not sure what you mean by “in the value” here The stakeholder group proposed that training facilitators in the value of open questioning could help
--	---

VERSION 2 – AUTHOR RESPONSE

Section	Comment	By	How it was addressed
Methods	Pg. 8 lines 28-44: In providing insight to the intervention and to the setting the authors provides some information to the population which is helpful but seems to suggest that CHTC is for married couples. However the cohabiting piece seems to be important. What is the cultural expectation regarding "marriage and relationships, are the authors using a western lens exclusively?	Reviewer 1	CHTC is not for married couples and we are sorry if the description of the intervention context was misleading in this way. We only report marriage and co-habiting rates as background contextual information on the community in which the intervention was implemented. These are not factors that we theorise directly impact on CHTC uptake, but are just important for the reader to be aware of in understanding the wider setting. As low marriage rates in KZN are already mentioned in para 2 of the introduction, and in order to make the context of our particular intervention clearer in the method, we have replaced the general background information about marriage rates in KZN with the actual marriage rates of our sample. We hope this helps ensure the context is clear, and is useful information for the reader: “The mean age was 26 years and 20% of

			the couples were married.”
Methods	Additionally there is a need to clarity on who has test and proportion of those who tested, who were diagnosed and who did not disclose. Regarding the intervention outcome, what proportion was testing for the first time.	Reviewer 1	We agree that these are important outcomes to include and these have been added to section 2.2: “The intervention was delivered in a community setting in KZN to 168 couples who had been in a relationship for at least six months. Just under two thirds of participants had had an HIV test before (63% of males and 60% of females) but only 20% of these had told their partner their HIV status. No couples had mutually disclosed their HIV status to each other. The mean age was 26 years and 20% of the couples were married”. And “Forty two percent of couples in the intervention group attended CHTC, of whom 46% were concordant HIV-negative, 30% were concordant HIV-positive, and 24% were serodiscordant (10). In 54% of the couples who took up CHTC in the intervention group, at least one partner was testing for the first time. Fifty nine percent of the participants who were diagnosed as HIV-positive were new diagnoses (10)”.
Results	An explanation of open questioning as a	Reviewer 1	We agree this was an important omission,

	strategy could be described a bit more.		thank you for noticing it. We have now added to section 3.1.2 that: “Open questioning involves asking the couple open-ended questions, such as “What makes it harder for you to test together?”, to enable the couple to discuss their barriers.”
Methods	There appears to have been several frameworks used and as written is a bit confusing as to the rationale and so some clarity around this upfront is warranted.	Reviewer 1	The table of changes from the person-based approach is used to inductively analyse the interview data. Meanwhile the logic model drew on existing theories (deductive) and the rationale for using the Theoretical Domains Framework (TDF) (Cane et al. 2012) and Capability Opportunity Motivation-Behaviour (COM-B) model is explained. We have sought to clarify the difference between the inductive and deductive use of frameworks in section 2.7: “The Table of Changes provides a technique for rapid inductive qualitative analysis” “This was a deductive process drawing on existing theory; the optimised intervention components were mapped on to constructs from the Theoretical Domains Framework (TDF) and Capability Opportunity Motivation-Behaviour (COM-B) model”

Results	The conclusion rounds out that more training is needed for counselors but the findings were almost exclusive to the couples. Were there any findings pertaining to the providers that can be shown in support of this. If so this could be incorporated.	Reviewer 1	We agree that there are limited data presented from the counsellors' perspective. This is due to the amount of data we wanted to include and the tight word count. However, we believe there are sufficient findings presented to support the conclusion about additional training for counsellors in open questioning skills. The quote from the staff member on page 14 supports this finding: “you speak and speak and speak and encouraging them about honesty and trust and everything but still tomorrow, no, I'm not ready to do the testing..... I kept on encouraging them that most of the people are testing negative, just to make them think about it (Staff member 1, counsellor and facilitator)” This suggests that this counsellor was trying to persuade and encourage couples to test, rather than asking them questions to help discuss their barriers.
All	Grammar and spelling needs to be assessed throughout.	Reviewer 1	Thank you, the paper has been re-read thoroughly and any grammatical or spelling errors corrected.
Methods	I do remain concerned about the 5 year gap, but the study does	Reviewer 2	Thank you. We agree the 5-year gap is a weakness of the study,

	appear to have achieved sound results that resonate with the community group.		but the point about the findings making sense to the community group has been added to the discussion: “The discussion with the community group helped confirm that the findings are relevant.”
Methods	It should be clarified that this study takes place within a community research site. This facilitates the ongoing follow-up of people and ensures service provision. Some greater context has been added, but could still do with more contextualisation. This is a qualitative and partly PAR paper, so context is essential.	Reviewer 2	We agree and this has been added to section 2.4: “This study took place within a community research site, which facilitated ongoing follow-up of people via contacts in the community, and generally helps ensure service provision.”
Methods	The method of sampling described is not purposive. It follows the outcomes of the study. Purposive implies a selection based in advance. Or is the purposive sampling based on a distribution across the 5 groups. This is not clear. Okay it is clarified later in the results. A line here in the methods making this clear would facilitate the reading of the manuscript.	Reviewer 2	This has been added to the methods (section 2.3) for clarity: “We sought to sample the same number of couples across the five groups.”
Methods	Pg 9 In 51 When couples were still in	Reviewer 2	Thank you, this change has been made.

	should read If couples		
Methods	The introduction of the authors background is still inadequate as it only cover the interviewers. All authors that are part of paper and analysis need to be introduced.	Reviewer 2	We would prefer not to add details of the background for every author into the paper, as this would be a lengthy section and we don't believe it is standard good practice for a qualitative paper. We feel that the description of the background of the interviewers and the lead data analyst (KM) is sufficient for understanding how their background might have influenced the findings.
Methods	The systems used to involve the stakeholders could be unpacked further. How was the analysis presented? Were any tools used to facilitate discussion? This is a novel system, so readers would benefit from hearing more. It would also allow for the methodology to be more easily replicated in other contexts.	Reviewer 2	Thank you for this helpful suggestion. A paragraph has been added to the PPI section to explain how the stakeholder workshops were run.
Results	Pg 15 In 24 I am not sure what you mean by "in the value" here The stakeholder group proposed that training facilitators in the value of open questioning could help	Reviewer 2	Thank you. We have now clarified what we mean in this sentence: "The stakeholder group proposed that training facilitators in the rationale for using open questioning could help encourage them to use this

			technique more often, and thereby enable a couple to discuss why they are not ready to test together and to come up with their own solutions.”
--	--	--	--

VERSION 3 – REVIEW

REVIEWER	Skinner, Donald Human Sciences Research Council, Faculty of Health Sciences
REVIEW RETURNED	22-Aug-2021

GENERAL COMMENTS	Paper should now be published. No further changes required.
---